# Wnt/β-Catenin Signaling Contributes to Paclitaxel Resistance in Bladder Cancer Cells with Cancer Stem Cell-Like Properties

**DOI:** 10.3390/ijms23010450

**Published:** 2021-12-31

**Authors:** Rocío Jiménez-Guerrero, Alejandro Belmonte-Fernández, M. Luz Flores, Mónica González-Moreno, Begoña Pérez-Valderrama, Francisco Romero, Miguel Á. Japón, Carmen Sáez

**Affiliations:** 1Instituto de Biomedicina de Sevilla (IBIS), Hospital Universitario Virgen del Rocío/CSIC/Universidad de Sevilla, 41013 Seville, Spain; rjimenez-ibis@us.es (R.J.-G.); mgonzalez-ibis@us.es (M.G.-M.); 2Department of Microbiology, Faculty of Biology, Universidad de Sevilla, 41012 Seville, Spain; abelmonte1@us.es (A.B.-F.); frport@us.es (F.R.); 3Department of Pathology, Hospital Universitario de Badajoz, 06080 Badajoz, Spain; marialuz.flores@salud-juntaex.es; 4Department of Oncology, Hospital Universitario Virgen del Rocío, 41013 Seville, Spain; bperez@gmail.com; 5Department of Pathology, Hospital Universitario Virgen del Rocío, 41013 Seville, Spain

**Keywords:** Wnt/β-catenin pathway, CSC phenotype, paclitaxel resistance, bladder cancer

## Abstract

The Wnt/β-catenin pathway plays an important role in tumor progression and chemotherapy resistance and seems to be essential for the maintenance of cancer stem cells (CSC) in several tumor types. However, the interplay of these factors has not been fully addressed in bladder cancer. Here, our goal was to analyze the role of the Wnt/β-catenin pathway in paclitaxel resistance and to study the therapeutic efficacy of its inhibition in bladder cancer cells, as well as to determine its influence in the maintenance of the CSC-like phenotype in bladder cancer. Our results show that paclitaxel-resistant HT1197 cells have hyperactivation of the Wnt/β-catenin pathway and increased CSC-like properties compared with paclitaxel-sensitive 5637 cells. Paclitaxel sensitivity diminishes in 5637 cells after β-catenin overexpression or when they are grown as tumorspheres, enriched for the CSC-like phenotype. Additionally, downregulation of β-catenin or inhibition with XAV939 sensitizes HT1197 cells to paclitaxel. Moreover, a subset of muscle-invasive bladder carcinomas shows aberrant expression of β-catenin that associates with positive expression of the CSC marker ALDH1A1. In conclusion, we demonstrate that Wnt/β-catenin signaling contributes to paclitaxel resistance in bladder cancer cells with CSC-like properties.

## 1. Introduction

Bladder cancer is the tenth most commonly diagnosed cancer worldwide, with approximately 573,000 new cases and 213,000 deaths in 2020 [1]. Although 70% are non-muscle-invasive bladder cancers, which have a good overall survival rate, 30% are muscle-invasive bladder cancers, which carry an increased risk of developing metastatic bladder cancer with a lower overall survival rate. Treatment for muscle-invasive bladder cancer most commonly involves radical surgery and chemotherapy. First-line chemotherapy for muscle-invasive bladder cancer and metastatic bladder cancer is cisplatin-based, but although many patients experience a significant initial response, they may eventually fail to the treatment. In this case, immunotherapy with checkpoint inhibitors is the preferred second-line alternative, but paclitaxel, docetaxel or vinflunine can be used as second- or later-line chemotherapy [2,3].

The Wnt/β-catenin pathway plays a critical role in embryonic development and tissue homeostasis in adult organisms. Given its importance in self-renewal and differentiation processes, it is not surprising that deregulation of the pathway leads to malignant proliferation, so that overexpression of Wnt and/or accumulation of β-catenin, mainly by inactivating mutations of tumor suppressor APC or by oncogenic mutations of β-catenin, have been described in many tumors that usually depend on Wnt/β-catenin for self-renewal or repair [4,5]. In the bladder, the Wnt/β-catenin pathway is necessary for the regeneration of the normal urothelium after injury, so its involvement in the pathogenesis of bladder cancer seems congruent. Indeed, it has been shown that β-catenin is overexpressed in human bladder cancer as compared to a normal urothelium and that Wnt/β-catenin signaling is activated and contributes to tumor progression [6,7,8]. Furthermore, a study of polymorphisms in 40 genes of the Wnt/β-catenin signaling pathway has suggested that variants may play a role in the etiology of bladder cancer [9]. With regard to therapy resistance, it has recently been documented that the Wnt/β-catenin pathway may be involved in resistance to cisplatin [10], doxorubicin [11,12], and immunotherapy [13] in bladder cancer. To our knowledge, its role in the acquisition of paclitaxel or docetaxel resistance has not been described. On the other hand, the role of the Wnt/β-catenin pathway has been described as essential for the maintenance of cancer stem cells (CSC) in several types of tumors, since it participates in key processes of these cells, such as self-renewal, dedifferentiation, inhibition of apoptosis or metastasis [14,15,16]. In support of this hypothesis is the observation that many markers used to identify and isolate CSC, such as the surface markers LGR5/GPR49, CD44, CD24 or EpCAM, or the proteins Oct-4 or ALDH1A1, are themselves Wnt/β-catenin target genes [5,17,18,19]. The activation of β-catenin occurs frequently in hematological tumors, allowing the establishment and drug resistance properties of CSC [20]. In addition, β-catenin has been shown to promote self-renewal and differentiation of prostate [21] and lung [22] CSC. In breast cancer, CSC have been shown to exhibit higher levels of Wnt/β-catenin signaling than the rest of the tumor, which appears to be associated with increased recurrence and worse prognosis [23,24]. However, although its involvement in malignant behavior has been proposed [25], the role of Wnt/β-catenin signaling in bladder CSC, as well as the therapeutic efficacy of its inhibition, are not yet fully addressed. Therefore, the main objective of this study was to analyze the involvement of the Wnt/β-catenin pathway in the acquisition of paclitaxel resistance, as well as to determine its role in the maintenance of the CSC phenotype in bladder cancer.

## 2. Results

### 2.1. HT1197 Cells Show Paclitaxel Resistance and Overexpress β-Catenin and CSC Markers as Compared with 5637 Cells

To study whether Wnt/β-catenin signaling may contribute to the paclitaxel response in bladder cancer, 5637 and HT1197 cells were treated with 0.1 µM paclitaxel for 48 h and then the induction of apoptosis and the levels of β-catenin and phospho-GSK3β^Ser9^ were analyzed by Western blot (Figure 1A). Apoptosis was induced in 5637 cells, as shown by open cleavage of PARP and expression of active caspase-3. HT1197 cells were resistant to paclitaxel, showing minor amounts of cleaved PARP or active caspase-3. In HT1197 cells, both total and active β-catenin levels were higher than in 5637 cells. Additionally, higher levels of β-catenin correlated with an increase in phospho-GSK3β^Ser9^, an inactive form of GSK3β^Ser9^ that prevents degradation of β-catenin. Moreover, in these cells, the levels of β-catenin (both in its total and active form) and phospho-GSK3β^Ser9^ were not decreased by paclitaxel, meaning that the signaling pathway is still active, so we believe that the maintenance of this signaling pathway could be contributing to the paclitaxel resistance observed. We next compared the expression of CSC markers in 5637 and HT1197 cells by Western blot (Figure 1B). Interestingly, HT1197 cells also expressed higher levels of ALDH1A1, Sox-2 and Oct-4 than 5637 cells, indicating an enrichment of the stem cell-like phenotype. To corroborate the presence of cell populations with stem cell-like properties, cell surface markers CD44 and CD24 were analyzed by flow cytometry (Figure 1C). Concordantly, CD44^+^/CD24^−^ events were more frequent in the HT1197 cell line, with 37.7% CD44^+^/CD24^−^ cells, than in the 5637 cell line, with only 1.9% CD44^+^/CD24^−^ cells. Taken together, these data suggest that the Wnt/β-catenin pathway plays a role in the paclitaxel resistance of bladder cancer cells with CSC-like properties.

### 2.2. Paclitaxel Sensitivity Diminishes in 5637 Cells after β-Catenin Overexpression or When Grown as Tumorspheres

To analyze the effect of increasing Wnt/β-catenin signaling in paclitaxel-sensitive 5637 cells, these were transiently transfected with the plasmid pCMVA-XL5-CTNNB1 and treated with 0.1 µM paclitaxel for 48 h (Figure 2). *CTNNB1* gene overexpression caused an increase in total and active β-catenin levels, as shown in Western blot. β-catenin-overexpressing 5637 cells showed lesser amounts of cleaved PARP and active caspase-3 after paclitaxel treatment, as compared with control cells, indicating decreased apoptotic cell death.

We also tested whether 5637 cells were able to acquire a more resistant phenotype when they were grown as tumorspheres. A paclitaxel-resistant cell line, 5637R, was used along with the parental 5637 cells. Figure 3A shows the absence of PARP cleavage and active caspase-3 in 5637R after paclitaxel treatment. We cultured both cell lines in non-adherent plates, either with RPMI 1640 or MammoCult^TM^, for 5 days (Figure 3B). Both 5637 and 5637R cells formed tumorspheres when using MammoCult^TM^ as the culture medium, but the number of tumorspheres was significantly higher in the latter. Moreover, 5637R cells were able to grow as tumorspheres even in RMPI 1640, so we can speculate that paclitaxel resistance is associated with the proliferation of tumorspheres. Interestingly, 5637 cells were less sensitive to paclitaxel when they were grown as tumorspheres in MammoCult^TM^ (Figure 3C). These tumorspheres of 5637 cells were also enriched in active β-catenin and the CSC markers ALDH1A1, Sox-2 and Oct-4 (Figure 3D). Taken together, these data show that conditions that favor the CSC-like phenotype also correlate with the activation of β-catenin and paclitaxel resistance.

### 2.3. Downregulation of β-Catenin or Inhibition with XAV939 Sensitize HT1197 Cells to Paclitaxel

We next addressed whether the interfering Wnt/β-catenin pathway could be a therapeutic strategy to reverse paclitaxel resistance. First, β-catenin was downregulated by siRNA in HT1197 cells, then these cells were treated with 0.1 µM paclitaxel for 48 h, and apoptotic cell death was analyzed by Western blot (Figure 4A). PARP cleavage and caspase-3 activation were seen upon the treatment, indicating that apoptosis was induced and that gene silencing of β-catenin restored paclitaxel sensitivity. Second, we treated HT1197 cells with 10 µM inhibitor XAV939 either alone or in combination with 0.1 µM paclitaxel for 48 h (Figure 4B). HT1197 cells did not die with XAV939 alone. In contrast, cell death was higher when the combination of paclitaxel and XAV939 was used, as shown by the significant increase in PARP cleavage and caspase-3 activation. XAV939 had no effect on total β-catenin levels, but the expression of the active form was significantly reduced, as well as after the combination treatment. This correlated with the decrease in phospho-GSK3β^Ser9^ after treatment with XAV939 alone or in combination with paclitaxel. Additionally, Oct-4 levels decreased after treatment, especially in the combination treatment, while ALDH1A1 and Sox-2 levels showed no significant differences. These results indicate that, although it does not induce cell death in these HT1197 cells, XAV939 was able to inhibit the Wnt/β-catenin pathway and promote apoptotic response when used in combination with paclitaxel. We also performed a viability assay with the *AlamarBlue Cell Viability Reagent* (Thermo Scientific, Waltham, MA, USA), and using Calcusyn software (Biosoft, Cambridge, UK, we were able to verify that paclitaxel plus XAV939 achieve synergy, with a combination index of 0.10685 (Appendix A).

### 2.4. A Subset of Muscle-Invasive Bladder Carcinomas Shows Aberrant Expression of β-Catenin and CSC Markers

Finally, we analyzed the expression of β-catenin, ALDH1A1 and Sox-2 by immunohistochemistry in tissue microarrays with biopsies from 140 patients with muscle-invasive bladder cancer (Figure 5). Tumors showed different immunostaining patterns for β-catenin. In most tumors (80.5%), β-catenin had an intense membrane-bound expression, or non-aberrant expression, indicating that the protein was inactive and linked to the degradation complex. In contrast, the other tumors (19.5%) showed nuclear or cytoplasmic expression or loss of membrane expression, so-called aberrant expression, indicating that the protein was active and able to exert transactivating actions. These results indicated that there was a small but significant group of muscle-invasive bladder cancer in which Wnt/β-catenin signaling pathway was active. Furthermore, we analyzed the expression of ALDH1A1 and Sox-2 in the same tissue microarrays. ALDH1A1 was positive in 23.3% of these muscle-invasive bladder cancer samples. There was a significant correlation between the pattern of β-catenin immunostaining and the expression of ALDH1A1. Only 17.8% of tumors with non-aberrant β-catenin expression were positive for ALDH1A1. In contrast, 38.5% of tumors with aberrant β-catenin expression showed ALDH1A1 immunostaining. Figure 5 shows one case with non-aberrant β-catenin, negative ALDH1A1 and Sox-2 (case 1) and one case with aberrant β-catenin, positive ALDH1A1 and Sox-2 (case 2). The contingency table (Table 1) shows the correlation between the number of tumors with aberrant or non-aberrant expression of β-catenin, and positive or negative ALDH1A1. This was statistically significant (*p*-value < 0.05). Therefore, we were able to conclude that β-catenin activation is related to the expression of CSC markers in muscle-invasive bladder cancer clinical samples, so that these markers could predict those patients who are more likely to progress after paclitaxel treatment.

## 3. Discussion

Taxanes are effective drugs for the treatment of solid tumors, including bladder cancer. Toxicity and drug resistance may limit their clinical benefits, so identifying markers that predict which patients respond to treatment is becoming increasingly important [26]. β-catenin is overexpressed in tumor samples as compared to normal urothelium, and its signaling contributes to tumor progression in bladder cancer [6,8]. In addition, we have shown that Wnt/β-catenin pathway is involved in paclitaxel resistance in prostate cancer [27], but this has not been addressed in bladder cancer. In this study, we have shown that the level of Wnt/β-catenin signaling is significantly higher in paclitaxel-resistant HT1197 cells, with higher levels of total and active β-catenin and phospho-GSK3β^Ser9^ than in paclitaxel-sensitive 5637 cells. Moreover, overexpression of β-catenin in the latter led to decreased paclitaxel-induced apoptosis, so we can say that the Wnt/β-catenin pathway might be involved in paclitaxel resistance in bladder cancer.

The Wnt/β-catenin pathway has been described as essential for the maintenance of CSC in several types of cancer [21,22,23,24,28]. Our results show that the activity of Wnt/β-catenin pathway was closely related with the CSC-like phenotype, as shown by the levels of CSC markers ALDH1A1, Oct-4 and Sox-2, which were higher in HT1197 than in 5637 cells. Some studies have described that β-catenin regulates the stemness phenotype through the control of some of its targets, such as ALDH1A1 [19] or Oct-4 [17], supporting this hypothesis. In addition, HT1197 cells also presented a major population of CD44^+^/CD24^−^ cells, one of the most widely accepted CSC-like properties [29], which together seem to indicate that these cells are enriched in the CSC-like phenotype. We also grew tumorspheres derived from parental 5637 and paclitaxel-resistant 5637R cells, as tumorspheres favor the overgrowth of cells with the CSC-like phenotype. Yoshida et al. showed that the Wnt/β-catenin pathway is necessary for the proliferation of bladder tumorspheres [25], but its relationship with paclitaxel resistance is unexplored. In our study, paclitaxel-resistant 5637R cells were able to form a higher number of tumorspheres than the parental 5637 cells when they were cultured in MammoCult^TM^. In fact, paclitaxel-resistant 5637R cells grew as tumorspheres even in non-adherent plates with RPMI 1640, demonstrating that paclitaxel resistance associates with the proliferation of bladder tumorspheres (and cells with the CSC-like phenotype). This could explain, at least in part, the recurrence and metastasis observed after chemotherapy. Accordingly, paclitaxel treatment induced cell death in lung cancer, but it was selected for ALDH^+^ CSC to promote metastasis in vivo [30]. Consistent with these data, paclitaxel-induced apoptosis in 5637 cell-derived tumorspheres was lower than in the same cells grown in adherent culture, supporting results which described that in vitro, breast tumorspheres are more resistant to paclitaxel than monolayer cultures [31]. Moreover, 5637 cell-derived tumorspheres were enriched in active β-catenin and the CSC markers ALDH1A1, Sox-2 and Oct-4, demonstrating that paclitaxel resistance promotes the proliferation of bladder tumorspheres through Wnt/β-catenin pathway.

It is widely accepted that tumors could be sustained by a small population of CSC resistant to conventional chemo- and radiotherapy. Therapies such as paclitaxel or cisplatin lead to a reduction in tumor bulk, but increase the proportion of CSC, generating recurrences and metastasis [32,33]. Our results suggest that paclitaxel resistance in bladder cancer could be overcome with therapeutic approaches targeting the Wnt/β-catenin pathway and its effects on CSC. Indeed, gene silencing of β-catenin in paclitaxel-resistant HT1197 cells resulted in increased apoptotic cell death. Some agents targeting CSC through inhibition of Wnt/β-catenin pathway are being evaluated in the treatment of urothelial bladder carcinoma [34,35,36]. Despite this, there are still few in vitro studies showing the therapeutic efficacy of inhibiting this pathway in combination with other conventional drugs, such as paclitaxel. Therefore, in this work, we used the inhibitor XAV939 in combination with paclitaxel to sensitize resistant bladder cancer cells. In this regard, we observed that combination treatment was able to decrease the levels of activation of the Wnt/β-catenin pathway and Oct-4, allowing the induction of apoptosis in resistant HT1197 cells. In support of our results, it has recently been shown that the combination of XAV939 and paclitaxel is able to inhibit tumor growth and metastasis in breast cancer [37]. Moreover, it has been demonstrated that other combinations of drugs with taxanes, such as salinomycin and docetaxel, are effective in eliminating both cancer cells and CSC in gastric cancer [38]. Thus, the concept of combining drugs that, together with conventional chemotherapy, are able to act both on tumor bulk and CSC, is strongly supported.

Finally, the expression of β-catenin and some of the CSC markers analyzed in our study, such as ALDH1A1, Sox-2 or Oct-4, correlates with tumor grade, recurrence, metastasis or worse survival in patients with bladder cancer [8,39,40,41,42]. In patients with muscle-invasive bladder cancer analyzed during this work, the expression of cytosolic or nuclear (and therefore active) β-catenin was associated with the positive expression of ALDH1A1. Although still preliminary studies, these results corroborate our data and emphasize the importance of their potential use as biomarkers in the clinic.

In summary, we demonstrate that the Wnt/β-catenin pathway promotes paclitaxel resistance and its involvement in the maintenance of the subpopulation of therapy-resistant CSC-like cells in bladder cancer. Therefore, β-catenin and CSC markers, such as ALDH1A1, could be of potential prognostic or predictive value in the treatment of patients with muscle-invasive bladder cancer.

## 4. Materials and Methods

### 4.1. Cell Culture and Drugs

Human 5637 bladder cancer cells were ordered from the Interlab Cell Line Collection (Genoa, Italy), and HT1197 bladder cancer cells from Sigma-Aldrich (St Louis, MO, USA). Both cell lines were derived from patients with muscle-invasive bladder carcinoma. The paclitaxel-resistant 5637R cell line was generated by treating 5637 cells with 4 nM paclitaxel for 2 months; then, the concentration of the drug was reduced by half. The surviving cells were maintained with 1 nM paclitaxel for 1 month and finally, the drug was withdrawn from the medium. The surviving cells were again subjected to the same treatment and the resulting paclitaxel-resistant cells were used for the assays. All experiments were performed using cells that had not exceeded the first ten passages after receipt and were routinely tested for mycoplasma contamination. 5637 and 5637R cells were cultured in RPMI-1640 (Lonza, Basel, Switzerland) supplemented with 10% fetal bovine serum (Sigma-Aldrich, St Louis, MO, USA), 50 U/mL penicillin (Sigma-Aldrich, St Louis, MO, USA), 50 mM streptomycin (Sigma-Aldrich, St Louis, MO, USA), 10 mM HEPES and 1 mM glutamine (Gibco, Waltham, MA, USA). HT1197 cells were cultured in MEM/EBSS (HyClone, Logan, UT, USA) supplemented with 10% fetal bovine serum (Biowest, Nuaillé, France), 50 U/mL penicillin, 50 mM streptomycin and 1 mM MEM non-essential amino acid solution (Sigma-Aldrich, St Louis, MO, USA). Cells were cultured at 37 °C in a humidified incubator under 5% CO_2_. Stock solutions of paclitaxel (Calbiochem, San Diego, CA, USA ) and XAV939 (Selleck Chemicals, Houston, TX, USA) were prepared at 10 mM in dimethyl sulfoxide (DMSO, Sigma-Aldrich, St Louis, MO, USA) and stored at −20 °C. In all experiments, cells were treated with either drug or vehicle during the log phase of growth. Cells were treated with 1 µM XAV939 or 0.1 µM paclitaxel either as a single treatment or in combination for 48 h.

### 4.2. Tumorsphere Formation Assay

For tumorsphere formation, cells were grown in MammoCult^TM^ Human Medium supplemented with MammoCult^TM^ Proliferation Supplement (StemCell Technologies, Vancouver, BC, Canada), 50 U/mL penicillin, 50 mM streptomycin, 4 µg/mL heparin (StemCell Technologies, Vancouver, BC, Canada) and 0.48 µg/mL hydrocortisone (StemCell Technologies, Vancouver, BC, Canada). Cells were plated in 24-well ultra-low attachment plates (Corning, Corning, NY, USA) at a density of 6000–20000 cells per well, depending on the cell line, and cultured for 5 days. To test the effect of paclitaxel, tumorspheres were treated with DMSO or 0.1 µM paclitaxel and they were collected after 48 h. The number of tumorspheres was quantified using an inverted microscope (Olympus, Tokyo, Japan).

### 4.3. Flow Cytometry Analysis of CD44 and CD24

To study the expression of CD44 and CD24, 10^6^ cells were resuspended in 125 µL of PBS with 12.5 µL of blocking buffer [1% Blocking Reagent (Roche, Mannheim, Germany) in 0.05% Tween 20-PBS] and incubated for 10 min on ice. Then, 5 µL of allophycocyanin (APC)-conjugated anti-CD44 antibody (Miltenyi, Bergisch Gladbach, Germany) and 5 µL of phycoerythrin (PE)-conjugated anti-CD24 antibody (Miltenyi, Bergisch Gladbach, Germany) were added and incubated for 30 min on ice, then two washes with PBS were performed. Cells were centrifuged for 10 min at 3000 rpm and finally resuspended in 500 µL of PBS for examination using a FACS Canto II analytical flow cytometer (BD Biosciences, San Jose, CA, USA). Samples without antibodies and single labeling with APC-CD44 or PE-CD24 were used as controls. The results were analyzed using Diva software (BD Biosciences, San Jose, CA, USA).

### 4.4. Small Interfering RNA (siRNA) and Plasmid Transfections

Validated ON-TARGETplus SMART pools of CTNNB1 (L-003482) and non-targeting (D-001810) siRNA, as a negative control, were obtained from GE Dharmacon (Lafayette, CO, USA). Transfections were carried out using DharmaFECT 2 reagent (GE Dharmacon, Lafayette, CO, USA) according to the manufacturer’s instructions. All siRNA pools were used at 50 nM. Cells were subjected to the different treatments 24 h after siRNA transfection. pCMV6-XL5 and pCMV6-XL5-CTNNB1 plasmids (Origene, Rockville, MD, USA) were transiently transfected using the FuGENE reagent (Promega, Madison, WI, USA), according to the manufacturer’s instructions. Cells were subjected to the different treatments 24 h after plasmid transfection.

### 4.5. Antibodies

Mouse monoclonal anti-PARP (1:500), anti-β-catenin (1:10,000) and anti-E-cadherin (1:4000) were from BD Biosciences (San Jose, CA, USA); mouse monoclonal anti-ALDH1A1 (1:500), rabbit polyclonal anti-Sox-2 (1:3000) and anti-Oct-4 (1:2000) were from Santa Cruz (Santa Cruz, CA, USA); mouse monoclonal anti-β-actin (1:20,000) was from Sigma-Aldrich (St Louis, MO, USA); rabbit polyclonal anti-cleaved caspase-3 (Asp175) (1:500) and anti-p-GSK3β^Ser9^ (1:1000) were from Cell Signaling (Danvers, MA, USA); and mouse monoclonal anti-active-β-catenin (1:500) was from Millipore (Burlington, MA, USA).

### 4.6. Western Blots

Cells were lysed in Nonidet P-40 (NP40) lysis buffer [10 mM Tris-HCl (pH 7.5), 150 mM NaCl, 10% glycerol and 1% NP40]. Equal amounts of total protein, as determined by the BCA protein assay kit (Pierce, Rockford, IL, USA), were subjected to SDS-PAGE on 8% polyacrylamide gels and transferred to Hybond ECL nitrocellulose membranes (GE Healthcare, Little Chalfont, UK). Blots were stained with Ponceau S to ensure protein amounts were equal. For immunodetection, blots were soaked in blocking buffer [1% Blocking Reagent (Roche, Mannheim, Germany) in 0.05% Tween 20-PBS] for 1 h and incubated with primary antibody in blocking buffer overnight at 4 °C. Blots were then washed in 0.05% Tween 20-PBS and incubated with either goat anti-mouse IgG (1:20,000, GE Healthcare, Little Chalfont, UK) or goat anti-rabbit IgG (1:20,000, GE Healthcare) peroxidase-labeled antibodies in blocking buffer for 1 h. An enhanced chemiluminiscent ECL system (GE Healthcare, Little Chalfont, UK) was applied according to the manufacturer’s protocol. All experiments were performed in triplicate. Scanning densitometry of blots was analyzed using QuantiScan software (Biosoft, Cambridge, UK).

### 4.7. Immunohistochemistry

Formalin-fixed, paraffin-embedded tissues from the transurethral resections of 140 patients with muscle-invasive bladder cancer were selected to make tissue microarrays using 1-mm-diameter tumor samples. The study was approved by the local ethical committee. Five-µm tissue sections were dewaxed, rehydrated and immersed in 3% H_2_O_2_ aqueous solution for 30 min to exhaust endogenous peroxidase. Heat-induced epitope retrieval was performed with 1 mM EDTA (pH 9.0) in a microwave oven. Sections were incubated overnight at 4 °C with anti-β-catenin (1:1000), anti-ALDH1A1 (1:1000) and anti-Sox-2 (1:200) antibodies. Peroxidase-labeled secondary antibody and 3,3ʹ-diaminobenzidine were applied according to the manufacturer’s protocol (Dako, Glostrup, Denmark). Slides were then counterstained with hematoxylin and mounted.

### 4.8. Statistical Analysis

Data comparing differences between two conditions were statistically analyzed, when indicated, using the paired Student’s *t* test. The association between the aberrant expression of β-catenin and low or high expression of ALDH1A1 was analyzed using Fisher’s exact test. Differences were considered significant when *p* < 0.05. Calculations were performed using Prism 6.0 (GraphPad, San Diego, CA, USA).

## Figures and Tables

**Figure 1 ijms-23-00450-f001:**
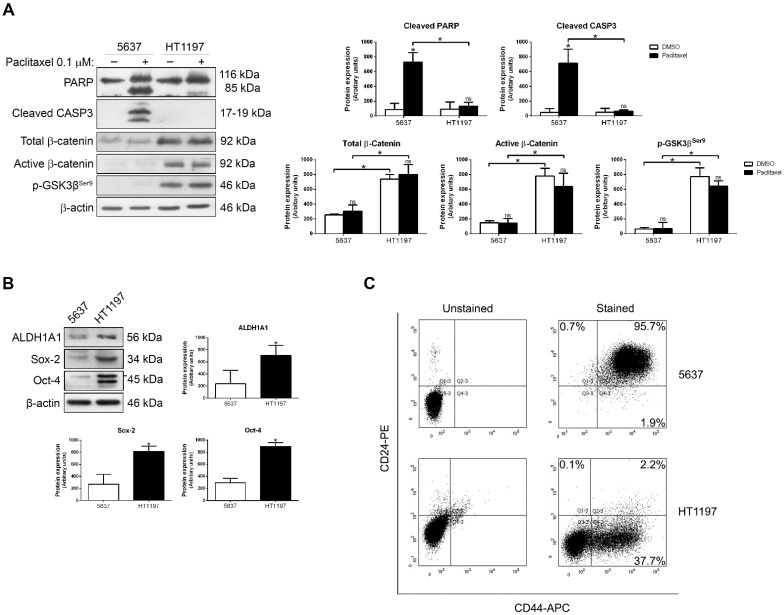
Paclitaxel-resistant HT1197 cells show β-catenin overexpression and CSC-like properties. (**A**) Western blots of PARP, cleaved caspase-3, total β-catenin, active β-catenin and p-GSK3β^Ser9^ in 5637 and HT1197 cells treated with DMSO (vehicle) or 0.1 µM paclitaxel for 48 h, and (**B**) Western blots of CSC markers ALDH1A1, Sox-2 and Oct-4 in untreated 5637 and HT1197 cells. β-actin was used as a loading control. Histograms show the densitometric analysis of the indicated proteins. Data are presented as mean ± SD. * *p* value < 0.05 from Student’s *t* test (n ≥ 3). ns, not significant. (**C**) Flow cytometry detection of CSC-like CD44^+^/CD24^−^ populations in 5637 and HT1197 cells. PE, phycoerythrin. APC, allophycocyanin.

**Figure 2 ijms-23-00450-f002:**
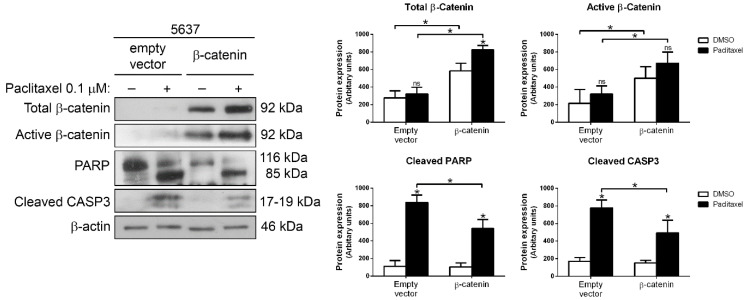
Overexpression of β-catenin reduces paclitaxel sensitivity in 5637 cells. Western blots of PARP, cleaved caspase-3, total β-catenin, and active β-catenin in 5637 cells transiently transfected with pCMV6-XL5-CTNNB1 or with empty vector and treated with DMSO (vehicle) or 0.1 µM paclitaxel for 48 h. β-actin was used as a loading control. Histograms show the densitometric analysis of the indicated proteins. Data are presented as mean ± SD. * *p* value < 0.05 from Student’s *t* test (n ≥ 3). ns, not significant.

**Figure 3 ijms-23-00450-f003:**
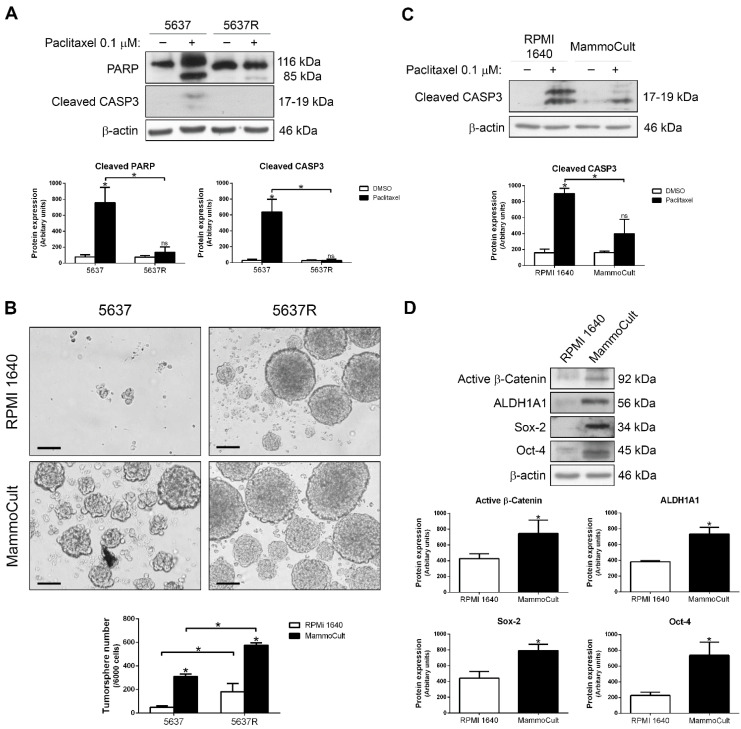
Paclitaxel sensitivity is reduced in 5637 cells grown as tumorspheres. (**A**) Western blots of PARP and cleaved caspase-3 in parental 5637 and paclitaxel-resistant 5637R cells treated with DMSO (vehicle) or 0.1 µM paclitaxel for 48 h. (**B**) Representative images of tumorspheres from parental 5637 and paclitaxel-resistant 5637R cells grown for 5 days in non-adherent conditions with RPMI 1640 or MammoCult^TM^. Bar, 100 µM. Histograms show the number of tumorspheres per well. (**C**) Western blot of cleaved caspase-3 in 5637 cells cultured in RPMI 1640 and adherent conditions or MammoCult^TM^ and non-adherent conditions and treated with DMSO (vehicle) or 0.1 µM paclitaxel for 48 h. (**D**) Western blots of active β-catenin, ALDH1A1, Sox-2 and Oct-4 in 5637 cells cultured in RPMI 1640 and adherent conditions or MammoCult^TM^ and non-adherent conditions. β-actin was used as a loading control. Histograms show the densitometric analysis of the indicated proteins. Data are presented as mean ± SD. * *p* value < 0.05 from Student’s *t* test (n ≥ 3). ns, not significant.

**Figure 4 ijms-23-00450-f004:**
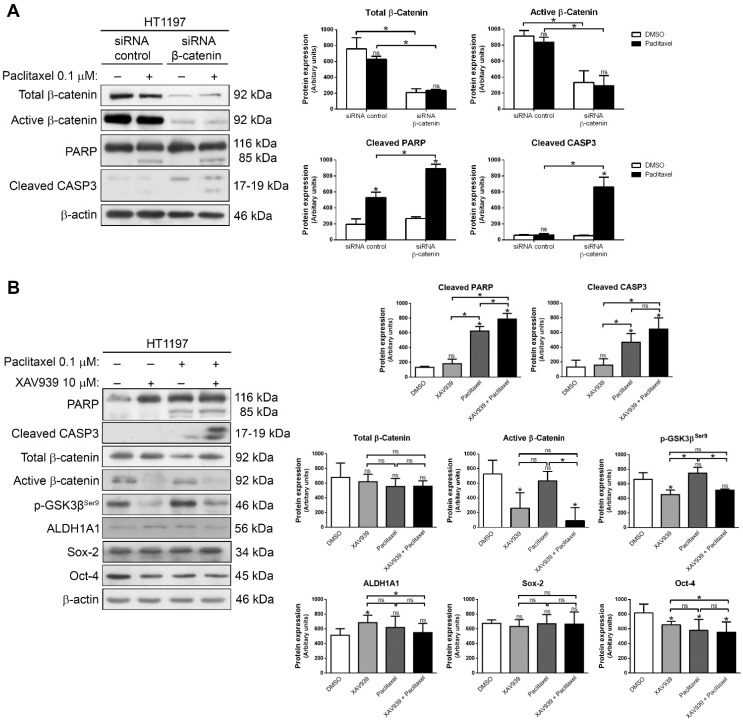
Interfering β-catenin sensitizes HT1197 cells to paclitaxel. (**A**) Western blots of PARP, cleaved caspase-3, total β-catenin, and active β-catenin in HT1197 cells transfected with siRNA β-catenin or non-targeting siRNA and treated with DMSO (vehicle) or 0.1 µM paclitaxel for 48 h, and (**B**) Western blots of PARP, cleaved caspase-3, total β-catenin, active β-catenin, p-GSK3β^Ser9^, ALDH1A1, Sox-2 and Oct-4 in HT1197 cells treated with DMSO (vehicle), 10 µM XAV939, 0.1 µM paclitaxel or 0.1 µM paclitaxel plus 10 µM XAV939 for 48 h. β-actin was used as a loading control. Histograms show the densitometric analysis of the indicated proteins. Data are presented as mean ± SD. * *p* value < 0.05 from Student’s *t* test (n ≥ 3). ns, not significant.

**Figure 5 ijms-23-00450-f005:**
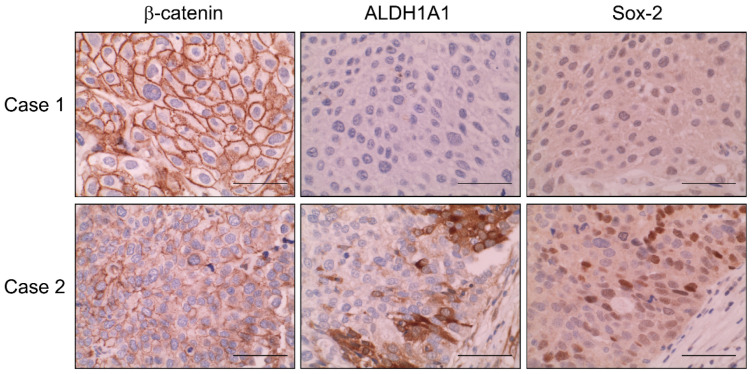
Expression of β-catenin and CSC markers in muscle-invasive bladder carcinomas. β-catenin, ALDH1A1, and Sox-2 immunostains of two representative cases of muscle-invasive bladder carcinomas are shown. Case 1 shows non-aberrant (membrane staining) β-catenin and absence of ALDH1A1 and Sox-2. Case 2 shows aberrant (loss of membrane staining) β-catenin and the presence of ALDH1A1 and Sox-2. Bar, 50 µM.

**Table 1 ijms-23-00450-t001:** Contingency table with the distribution of β-catenin and ALDH1A1 immunostains in 133 cases of muscle-invasive bladder carcinomas.

		β-Catenin Expression	
		Not Aberrant (Inactive)	Aberrant (Active)
ALDH1A1 expression	Negative	88	16
	Positive	19	10

*p* value = 0.0328 from Fisher’s exact test.

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
