# Peer review of "Wnt/β-Catenin Signaling Contributes to Paclitaxel Resistance in Bladder Cancer Cells with Cancer Stem Cell-Like Properties"

_ijms, 2021, doi:10.3390/ijms23010450_

Round 1

Reviewer 1 Report

To the authors of the manuscript: “Wnt / β-catenin signaling contributes to paclitaxel resistance in

bladder cancer cells with cancer stem cell-like properties ”.

The article has seemed to me of great relevance for the prognosis of therapy in patients with bladder cancer, especially in those with few options that recur and metastasize. The results offered by their research shed light on the involvement of this signaling pathway in these tumors and their behavior in drug resistance. Certainly a promising job. However, precisely because I value the quality of this work, I think that adding some experiments, it would be an article of the highest quality and interest. Also, I would like to discuss some points that I find interesting.

Q1. I would like to know how the two cell lines have been chosen for the assays (HT1197, 5637). For example, the HT1197 cell line derived from a grade 4 transitional cell carcinoma of the bladder of a 44 year old caucasian male who had not received chemo or radiation therapy, so how was its resistance to paclitaxel known? What is known about the mutational landscape of these cell lines? Both lines are MIBC, it should be included in materials and methods. Have the treatments been tested in some high-grade NMIBC cell line? It would be interesting, especially since this type of tumors share many characteristics with MIBCs, and they are usually the prelude.

Q2. I'd like to see trials with at least one other shRNA.

Q3. All drug combination trials show wb and quantification. I think it would be interesting to carry out an analysis where the combination index that determines synergy, addition or antagonism could be seen.

Q4. The tumorosphere assays are based on the resistant 5637 line. Why was the line with overexpression of the CTNNB1 gene not included in these tests? It would be a second way to demonstrate the involvement of the pathway, first by generating resistance to the drug for months, and second by directly overexpressing β-catenin in cells. In addition, it should be taken into account that during resistance tests for months, cultures have always been in the presence of antibiotics, a widespread practice that can mask bacterial action.

Q5. I understand that the inactive form of GSK3β, prevents its inhibitory action on β-catenin and it cannot be degraded. In addition to the role that this signaling plays in resistance. But some studies showing the STAT3 implication in MIBC, and studies showing a crosstalk between the STAT3 and Wnt / β-catenin signaling pathways, where an activation of STAT3 in turn promotes the activation of Wnt and this would act on the levels of β-catenin. It would be interesting to look through Western blot at how the expression of other oncogenic signaling pathways is found, such as total and phosphorylated STAT3, as well as some epigenetic enzymes. In the case of Wnt/β-catenin pathway, its activation through EZH2 in BC cells proliferation has been already described, but the exact activation mechanism is not known. In invasive muscle tumors it is known that at least 89% of chromatin remodelers are altered. EZH2 is one of the main enzymes that controls gene repression and has other “non-canonical” functions. It plays a fundamental role in the recurrence and progression of bladder cancer, as well as in the maintenance of stem cells, both in development and in CSCs. Its expression is also known to be related to resistance to drugs, such as cisplatin in BC. EZH2 can act as scaffold protein for various transcriptional factors, such as estrogen receptors (ER) or components of the Wnt/ β-catenin signaling pathways to promote gene transcription. For example, EZH2 can activate the non-canonical Wnt signaling through the repression of DKK-1, an inhibitor of the Wnt co-receptor LRP in lung cancer. miR-144 downregulation increases bladder cancer cell proliferation by targeting EZH2 and regulating Wnt signaling. These are simple examples.

With RNA samples from the tests in this work, genomics and transcriptomics studies could be carried out that would help find genes that control, genes that are expressed, genes that are repressed, involved transcription factors, etc.

I close the part of in vitro tests with these questions. As the authors say, there are few in vitro studies that demonstrate therapeutic efficacy of inhibiting this pathway in combination with other drugs such as paclitaxel. Therefore, for them some more tests are necessary that would give robustness to this study.

In a second section, I would like to talk about patients.

Q1. Have only patients' TMAs been accessed for staining? What information is available?. Within materials and methods, only one sentence from the patient samples comments: “from the transurethral resections of 140 patients with muscle-invasive bladder cancer were selected to make tissue microarrays using 1-mm-diameter tumor samples”. Have these patients been treated with chemotherapy? Have they been treated with paclitaxel and have patients been resistant?

Q2. It would be very important to have access to RNA seq data, which would allow studying the mechanisms by which wnt / catenin is related to CSCs, as well as the study of public data in support. Genomic and transcriptomic analysis would open a wide range of possibilities and answers.

Q3. In addition, there are patient samples that present CSC marker expression but without active catenin expression, therefore the identification of these patients would be interesting.

Finally, I would like to emphasize that the work has a good focus, but it needs robustness, because it has potential and future publications could be of great impact.

Reviewer 2 Report

Thank you for submitting your paper. 

I have some comments for your paper. 

1) There may be misunderstandings, so it would be better to clarify.
Are you saying that the results in Figure 1A are the same before and after paclitaxel treatment?

2) Although in vitro studies were well conducted, it would be better to add the results of cytotoxicity assay and migration assay, which are basic analyses.

3) Is there any particular reason for not conducting animal studies? If you can add the results of animal studies, you will be able to a clearer conclusion.

4) As your introduction, paclitaxel can be used as 3rd line chemotherapies, but in most real situation, most patients are considered to participate in clinical trials before using paclitaxel. Moreover, as you commented,  it has already been documented that Wnt/β-catenin pathway may be involved in resistance to cisplatin, doxorubicin, and immunotherapy in bladder cancer. So your conclusion that β-catenin and CSC markers, such as ALDH1A1, could be of potential prognostic or predictive value in the treatment of patients with muscle-invasive bladder cancer may be not interesting and new to most clinicians.

Round 2

Reviewer 2 Report

Thank you for your proper revision.